

# Equilibration of quantum cat states

**Tony Jin[1,2]***

**1** DQMP, University of Geneva, 24 Quai Ernest-Ansermet, CH-1211 Geneva, Switzerland
**2** Laboratoire de Physique de l'École Normale Supérieure, CNRS, ENS & PSL University, Sorbonne Université, Université de Paris, 75005 Paris, France

* zizhuo.jin@unige.ch

## Abstract

We study the equilibration properties of isolated ergodic quantum systems initially prepared in a *cat state,* i.e a macroscopic quantum superposition of states. Our main result consists in showing that, even though decoherence is at work in the mean, there exists a *remnant* of the initial quantum coherences visible in the strength of the fluctuations of the steady state. We back-up our analysis with numerical results obtained on the XXX spin chain with a random field along the z-axis in the ergodic regime and find good qualitative and quantitative agreement with the theory. We also present and discuss a framework where equilibrium quantities can be computed from general statistical ensembles without relying on microscopic details about the initial state, akin to the eigenstate thermalization hypothesis.

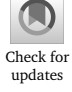

# 1 Introduction

Upon encountering the quantum statistical ensembles for the first time, one is often struck by the strong similitude they share with their classical counterpart. Indeed, quantum ensembles such as e.g the Gibbs ensemble, appear like a mere transcription of classical ones where one would have replaced the possible classical configurations by the eigenstates of the Hamiltonian. An explanation dating back to the early days of quantum mechanics [1, 2] is that, assuming ergodicity, the off-diagonal elements undergo a dephasing that time-averages to zero given that the different frequencies of the Hamiltonian are incommensurate. Thus, in the mean steady-state, purely quantum mechanical features such as superposition of state and entanglement are lost : this is a *decoherence* effect. Furthermore, the previous years have seen the development of a general framework known as the *eigenstate thermalization hypothesis* (ETH) which explains the emergence of statistical ensembles from a given set of assumptions on the spectral properties of the observables of the system [3–6] . The validity or invalidity of the ETH has been tested numerically in a certain number of studies [6–10].

Therefore, one could legitimately ask what is the consequences of having purely quantum features such as superposition of states and entanglement in the initial state of the system on the final equilibrium properties, if there are any at all? In this work, we intend to prove that, even if on average information about the quantum coherence of the initial state is lost at equilibrium, there is a remnant of the latter visible in the *fluctuations* around the stationary state. This phenomenon was already seen in a model of stochastic fermionic chain on a discrete lattice [11,12] and we provide here the generalization of these results to *any* ergodic quantum system.

In the context of ETH, one important assumption is that the initial states considered must have an energy comprised in a narrow energy shell. This assumption is tightly bound with having initial states which fulfill a *cluster decomposition* [13] constraint, i.e that the typical coherence length is small compared to the size of the system. Within this hypothesis, the fluctuations of the state around its average value scale like the inverse of the dimension of the Hilbert space and are thus exponentially suppressed as one increases the system size [14].

We are interested in situations where these hypothesis are relaxed, i.e for which the initial state of the system can be a superposition of states which have energies largely spread across the spectrum or equivalently that entangle large part of the system together. This typically happens for *cat states* which are quantum superposition of macroscopically distinct states -see fig.1. Cat states have attracted humongous interest from the physics community in the recent years [15–20] as their creation, stabilization and manipulation constitute key steps towards quantum computing and simulation. Working in the Hamiltonian basis, we will see that such states present non trivial, possibly non-local fluctuations of the off-diagonal components in the steady-state that are fixed by the initial quantum coherences.

This paper is organized as follows : First, we introduce our definition for quantum ergodicity and following it, compute first and second order correlation functions for elements of the density matrix. In a second part, we show that these results are in qualitative and quantitative agreement with numerical results obtained in a quantum ergodic spin chain. We then discuss a more general framework where the equilibrium ensembles don't depend on the fine structure of the initial states and discuss connection with ETH. We finally end by some concluding remarks and perspectives.

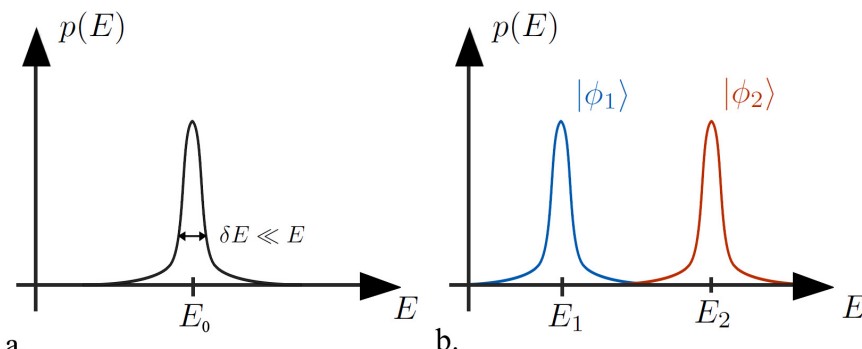

Figure 1: a. Traditional situation where the set of initial states all belong to a narrow energy window centered around $E_0$. b. Typical situation we will consider in this paper where we have a cat state made of a quantum superposition $1/\sqrt{2}(|\Phi_1\rangle + |\Phi_2\rangle)$

## 2 Ergodic hypothesis and equilibrium state

In classical physics, ergodicity is the hypothesis that at long-times, when the system reaches equilibrium, there is an equivalence between the time average of quantities and an ensemble average over a *microcanonical* distribution. The microcanonical distribution stipulates that, at fixed energy for an isolated system, the probability of all microscopic configurations are equal. Physically, the equivalence between the two averages comes from the assumption that at long-time the system explores isotropically all the degrees of freedom available under the constraint of fixed energy.

This work is devoted to the formulation and study of a similar *quantum ergodic hypothesis* : Let $\rho_0$ be the density matrix containing information about the initial conditions of the system. Given a set of conserved observables, $\hat{H}_1, \hat{H}_2, \cdots \hat{H}_n$, an accessible state is defined as a density matrix $\rho$ such that there exists a unitary $U$ commuting with all the $\hat{H}_i$ and fulfilling $U\rho_0 U^\dagger = \rho$. The quantum ergodic hypothesis asserts that in the long-time equilibrium state, all accessible density matrices have same probability weight or equivalently, that time average of elements of $\rho_t$ is equivalent to ensemble average over all possible unitary evolution $U$ that commutes with the conserved quantities. For simplification, in this paper, we will consider a unique conserved quantity $\hat{H}$ but generalization to a set of mutually commuting observables is straightforward.

Let us introduce some notations. We call $\mathcal{G}$ the group formed by all the unitaries such that $[U, \hat{H}] = 0$. We call $|E_i^{\nu_i}\rangle$ the eigenvector corresponding to energy $E_i$ for $\hat{H}$ where $\nu_i$ is an index accounting for possible degeneracies. We will call $d_i$ the dimension of the subspace associated to energy $E_i$. Because of the commutativity of the $U$'s with $H$, the group $\mathcal{G}$ can be decomposed as a direct product of $SU(d_i)$ : $\mathcal{G} = \times_i SU(d_i)$. Alternatively, this means that in the eigenbasis of the Hamiltonian, $U$ can be written in blocks indexed by $i$ with an element $U^{(i)}$ of $SU(d_i)$ in each block. This constitutes a *fundamental representation* of $\mathcal{G}$. We also introduce the decomposition of $\rho_0$ into different sectors $\rho_0^{(i,j)}$ defined as $\rho_0^{(ij)} = \sum_{\nu_i, \nu_j} \mathrm{tr}(\rho_0 |E_i^{\nu_i}\rangle \langle E_j^{\nu_j}|) |E_i^{\nu_i}\rangle \langle E_j^{\nu_j}|$

and $\rho_0 = \sum_{i,j} \rho_0^{(i,j)}$.

$$\rho_0 = \begin{pmatrix} \underbrace{\left(\rho^{(11)}\right)}_{d_1 \times d_1} & \underbrace{\left(\quad \rho^{(12)}\quad\right)}_{d_2 \times d_1} & \cdots \\ \underbrace{\left(\rho^{(21)}\right)}_{d_1 \times d_2} & \underbrace{\left(\quad \rho^{(2,2)}\quad\right)}_{d_2 \times d_2} & \\ \vdots & & \ddots \end{pmatrix}.$$

We will now illustrate how our quantum ergodic hypothesis allow to compute equilibrium quantities. We begin by considering the *average* of $\rho$ denoted by $\mathbb{E}[\rho_0]$ with respect to the ensemble average we just introduced. By definition :

$$\mathbb{E}[\rho_0] = \int_{\mathcal{G}} d\eta(U) U \rho_0 U^\dagger. \tag{1}$$

$\eta$ is the natural measure on $\mathcal{G}$, that is $d\eta(U) = \prod_i d\eta^{(i)}(U^{(i)})$ with $d\eta^{(i)}$ the Haar measure, i.e the unique invariant measure on $SU(d_i)$. The physical interpretation of this expression is exactly the one we discussed before : The average evolution is given by summing over all possible evolutions that preserve the spectrum of the Hamiltonian with the probability weight of each of them distributed uniformly with respect to the Haar measure. Making use of the decomposition in sectors of $\rho_0$, we have :

$$\begin{aligned} \mathbb{E}[\rho_0] &= \sum_i \mathbb{E}[\rho_0^{(i,i)}] + \sum_{i \neq j} \mathbb{E}[\rho_0^{(i,j)}] \tag{2} \\ &= \sum_i \int_{\mathcal{G}} d\eta^{(i)}(U^{(i)}) U^{(i)} \rho_0^{(i,i)} U^{(i)\dagger} + \sum_{i \neq j} \int_{\mathcal{G}} d\eta^{(i)}(U^{(i)}) d\eta^{(j)}(U^{(j)}) U^{(i)} \rho_0^{(i,j)} U^{(j)\dagger}. \tag{3} \end{aligned}$$

By the left invariance of the Haar measure we have that for $i \neq j$, $\forall U^{(i)} \in SU(d_i)$, $U^{(i)} \mathbb{E}[\rho_0^{(i,j)}] = \mathbb{E}[\rho_0^{(i,j)}]$ which is only true if $\mathbb{E}[\rho_0^{(i,j)}] = 0$. For $i = j$, Schur lemma tells us that $\mathbb{E}[\rho_0^{(i,j)}]$ must be proportional to the identity. The proportionality coefficient is determined by taking the trace so that we get :

$$\mathbb{E}[\rho_0] = \sum_i \frac{1}{d_i} \mathbb{I}^{(ii)} \text{tr}(\rho_0^{(ii)}). \tag{4}$$

Thus, in average, as one expects from decoherence, information about "off-diagonal" correlations between different energy sectors is lost. However, we will see in what follows that there is actually a remnant of the latter when one goes to higher order correlations.

Before going on, let's notice two extreme cases of interest : the first case is when all the energy levels are non-degenerate. Then, $\mathbb{E}[\rho_0]$ is just the diagonal ensemble, i.e the density matrix in which one has set all the off-diagonal components to zero. The second case is when there is only one energy sector that is the whole Hilbert space itself. We then have $\mathbb{E}[\rho_0] = \frac{1}{d}\mathbb{I}$. In this case, the density matrix states tells us all states with the same energy $E$ have the same probability weight, i.e it is the microcanonical ensemble. Fully-degenerate spectrum corresponds in general to chaotic or non-integrable systems, so one should expect that the diagonal ensemble describes accurately the steady state of such systems [14]. However, in practice, we know that equilibrium states of isolated system are accurately described by the microcanonical ensemble which corresponds to the steady-state of a fully degenerate spectrum. To go from the first ensemble to the second is not a trivial task which requires additional assumptions. We will discuss this point in more details in the section 4.

The second moment of elements of the density matrix is by definition :

$$\mathbb{E}[\rho_0^{\otimes 2}] = \int_{\mathcal{G}} d\eta(U) U^{\otimes 2} \rho_0^{\otimes 2} U^{\dagger \otimes 2}, \tag{5}$$

with $X^{\otimes n} \equiv \underbrace{X \otimes \cdots \otimes X}_{n \text{ times}}$. This quantity can be computed by generalizing arguments used for the mean. Again, it relies on the decomposition of $\rho_0^{\otimes 2}$ into sectors $(\rho_0^{\otimes 2})^{(i_1, j_1, i_2, j_2)} \equiv \sum_{\nu_{i_1}, \nu_{i_2}, \nu_{j_1}, \nu_{j_2}} \mathrm{tr}(\rho_0^{\otimes 2} |E_{i_1}^{\nu_{i_1}}, E_{i_2}^{\nu_{i_2}}\rangle \langle E_{j_1}^{\nu_{j_1}}, E_{j_2}^{\nu_{j_2}}|) |E_{i_1}^{\nu_{i_1}}, E_{i_2}^{\nu_{i_2}}\rangle \langle E_{j_1}^{\nu_{j_1}}, E_{j_2}^{\nu_{j_2}}|$ and identifying the invariant objects under the action of $U \otimes U$. We simply state the result and present the full derivation in app.A:

$$
\begin{aligned}
\mathbb{E}[\rho_0^{\otimes 2}] = & \sum_{i_1} \frac{1}{d_{i_1}(d_{i_1}+1)} ((\mathrm{tr}(\rho_0^{(i_1)}))^2 + \mathrm{tr}((\rho_0^{(i_1)})^2)) \mathbb{I}_{(2)}^{(i_1, i_1, i_1, i_1)} \\
& + \frac{1}{d_{i_1}(d_{i_1}-1)} ((\mathrm{tr}(\rho_0^{(i_1)}))^2 - \mathrm{tr}((\rho_0^{(i_1)})^2)) \mathbb{I}_{(1,1)}^{(i_1, i_1, i_1, i_1)} \\
& + \sum_{i_1 \neq i_2} \frac{1}{d_{i_1} d_{i_2}} (\mathrm{tr}(\rho_0^{(i_1 i_1)}) \mathrm{tr}(\rho_0^{(i_2 i_2)}) \mathbb{I}^{(i_1 i_2 i_1 i_2)} + \mathrm{tr}(\rho_0^{(i_1 i_2)} \rho_0^{(i_2 i_1)}) \mathbb{I}^{(i_1 i_2 i_2 i_1)}),
\end{aligned}
\tag{6}
$$

where the different identities are defined as :

$$\mathbb{I}_{(2)}^{(i_1, i_1, i_1, i_1)} = \sum_{\nu_{i_1}, \nu'_{i_1}} \frac{1}{4} (|E_{i_1}^{\nu_{i_1}}, E_{i_1}^{\nu'_{i_1}}\rangle + |E_{i_1}^{\nu'_{i_1}}, E_{i_1}^{\nu_{i_1}}\rangle)(\langle E_{i_1}^{\nu_{i_1}}, E_{i_1}^{\nu'_{i_1}}| + \langle E_{i_1}^{\nu'_{i_1}}, E_{i_1}^{\nu_{i_1}}|) \tag{7}$$

$$\mathbb{I}_{(1,1)}^{(i_1, i_1, i_1, i_1)} = \sum_{\nu_{i_1} \neq \nu'_{i_1}} \frac{1}{4} (|E_{i_1}^{\nu_{i_1}}, E_{i_1}^{\nu'_{i_1}}\rangle - |E_{i_1}^{\nu'_{i_1}}, E_{i_1}^{\nu_{i_1}}\rangle)(\langle E_{i_1}^{\nu_{i_1}}, E_{i_1}^{\nu'_{i_1}}| - \langle E_{i_1}^{\nu'_{i_1}}, E_{i_1}^{\nu_{i_1}}|) \tag{8}$$

$$\mathbb{I}^{(i_1, i_2, i_1, i_2)} = \sum_{\nu_{i_1}, \nu_{i_2}} |E_{i_1}^{\nu_{i_1}}, E_{i_2}^{\nu_{i_2}}\rangle \langle E_{i_1}^{\nu_{i_1}}, E_{i_2}^{\nu_{i_2}}|, \tag{9}$$

$$\mathbb{I}^{(i_1, i_2, i_2, i_1)} = \sum_{\nu_{i_1}, \nu_{i_2}} |E_{i_1}^{\nu_{i_1}}, E_{i_2}^{\nu_{i_2}}\rangle \langle E_{i_2}^{\nu_{i_2}}, E_{i_1}^{\nu_{i_1}}|. \tag{10}$$

The subscripts $(2)$ and $(1,1)$ refers respectively to the symmetric and antisymmetric irreducible representations of $SU(d_i) \otimes SU(d_i)$. The important point is that contrary to (4), (6) contains information about quantum superposition of states both in the case where they belong to the same sector (line 1 of (6)) but also when the superposition involves states belonging to different sectors (line 3 of (6)). Thus, two initial states having the same diagonal elements may relax to the same density matrix in average but present differences at the level of fluctuating quantities, providing a signature of the presence or the absence of initial quantum coherences.

One can carry this procedure to get access to higher moments but their explicit expression becomes more and more involved. We present the general formula in the app.B.

We will now illustrate these ideas on a concrete numerical example.

## 3 Numerical results on the XXX spin chain with random field in the ergodic regime

We test the predictive power of our model on the XXX model with random local fields :

$$\hat{H} = \sum_{j=0}^{L-2} J \vec{\sigma}_j \cdot \vec{\sigma}_{j+1} + \sum_{j=0}^{L-1} h_j \sigma_j^z, \tag{11}$$

with $L$ the lattice size and $\sigma_j^a$ the usual Pauli matrices. The boundaries are open. The $h_j$ are independent random variables distributed uniformly in an interval $[-h, h]$. The transition from an ergodic to a localized regime of this model has been studied in [21] and characterized by the spectral properties of $\hat{H}$. A quantity of particular interest is the mean ratio of consecutive level spacings known to be close to the one of the Wigner distribution ($\approx 0.53$) in the ergodic regime and to the one of the Poisson distribution ($\approx 0.38$) in the localized regime. For $J = 1$ and lattice size ranging from 11 to 22, it has bee shown in [21] that the transition between the two regimes occurred for $h \approx 2.5$. Since we are interested in the ergodic regime we will fix the value of $h$ to 1. We work in the minimal magnetization sector, i.e 0 for $L$ even and 1 for $L$ odd.

Let $\hat{O}$ be an observable. We will compute the time-evolution of $O(t) \equiv \mathrm{tr}(\rho_t \hat{O})$ by using exact diagonalization methods [22–24]. We denote the time-average by $\mathbb{E}_t[\bullet] \equiv \lim_{T \to \infty} \frac{1}{T} \int_{t=0}^{T} \bullet \, dt$ and will be interested in first and second order correlation functions $\mathbb{E}_t[O(t)]$, $\mathbb{E}_t[O(t)O'(t)]$. Our point will be to show that the time average $\mathbb{E}_t[\bullet]$ is equivalent to the previously introduced ensemble average over possible unitary evolutions $\mathbb{E}[\bullet]$.

We will study a *quench* situation in which the initial state is expressed in terms of eigenvalues of the Hamiltonian $\hat{H}$ from which we remove the XXX coupling between sites $L/2-1$ and $L/2$ (suppose $L$ even for simplicity), i.e $\hat{H}_0 = \hat{H}_{\mathrm{L}} + \hat{H}_{\mathrm{R}}$ with $\hat{H}_{\mathrm{L}} \equiv \sum_{j=0}^{L/2-2} J\vec{\sigma}_j \cdot \vec{\sigma}_{j+1} + \sum_{j=0}^{L/2-1} h_j \sigma_j^z$ and $\hat{H}_{\mathrm{R}} \equiv \sum_{j=L/2}^{L-2} J\vec{\sigma}_j \cdot \vec{\sigma}_{j+1} + \sum_{j=L/2}^{L-1} h_j \sigma_j^z$. The initial states chosen this way have well-defined energies $E_{\mathrm{R}}$ and $\mathrm{E}_{\mathrm{L}}$ and will be denoted $|E_{\mathrm{R}}, E_{\mathrm{L}}\rangle$. At time $t = 0$, we switch the Hamiltonian from $\hat{H}_0$ to $\hat{H}$, so that the system is now in an out-of-equilibrium situation.

To illustrate the importance of the presence or absence of initial quantum coherences in the steady state of the system, we propose to study two different set of initial conditions. They will be both indistinguishable from the point of view of their mean energy but they will encode for different off-diagonal quantum coherences which effect will be visible in the equilibrium fluctuations of the system. Let $E_{\min}$, $E_{\max}$ be the minimum and maximum energy of the spectrum and $|\Phi_1\rangle \equiv |E_{\mathrm{R}}^1, E_{\mathrm{L}}^1\rangle$, $|\Phi_2\rangle = |E_{\mathrm{R}}^2, E_{\mathrm{L}}^2\rangle$ such that $E_{\mathrm{R}}^1 + E_{\mathrm{L}}^1$ is close to $E_{\min}$ and $E_{\mathrm{R}}^2 + E_{\mathrm{L}}^2$ is close to $E_{\max}$. The decomposition of these states in the eigenbasis of $\hat{H}$ are shown in the app.C.

In the *protocol I*, corresponding to a *cat state* made of a quantum superposition of two states with «macroscopically» distinct energies $E_{\mathrm{R}}^1 + E_{\mathrm{L}}^1$ and $E_{\mathrm{R}}^2 + E_{\mathrm{L}}^2$, the initial state is chosen to be :

$$|\psi_0^{\mathrm{I}}\rangle = \frac{1}{\sqrt{2}}(|\Phi_1\rangle + |\Phi_2\rangle). \tag{12}$$

In the *protocol II*, corresponding to a *mixed state*, the initial state is described by the density matrix :

$$\rho_0^{\mathrm{II}} = \frac{1}{2}(|\Phi_1\rangle\langle\Phi_1| + |\Phi_2\rangle\langle\Phi_2|). \tag{13}$$

In both protocols, the reduced density matrices on R and L are the same.

We compute the time-evolution of two observables : $H_{\mathrm{R}}(t) \equiv \mathrm{tr}(\rho_t \hat{H}_{\mathrm{R}})$ and $Q(t) = \mathrm{tr}(\rho_t \hat{Q})$, $\hat{Q} \equiv |\Phi_1\rangle\langle\Phi_2| + |\Phi_2\rangle\langle\Phi_1|$. Note that $Q$ is non-local, in the sense that it has non zero support on the whole physical space. From formula (4,6) we can deduce the predictions for first and second moments of these quantities given by ensemble averages in both protocols. Importantly we have that :

$$\mathbb{E}^{\mathrm{I}}[H_{\mathrm{R}}] = \mathbb{E}^{\mathrm{II}}[H_{\mathrm{R}}], \quad \mathbb{E}^{\mathrm{I}}[H_{\mathrm{R}}^2] = \mathbb{E}^{\mathrm{II}}[H_{\mathrm{R}}^2], \tag{14}$$

$$\mathbb{E}^{\mathrm{I}}[Q] = \mathbb{E}^{\mathrm{II}}[Q] = 0, \quad \mathbb{E}^{\mathrm{I}}[|Q|^2] \neq \mathbb{E}^{\mathrm{II}}[|Q|^2], \tag{15}$$

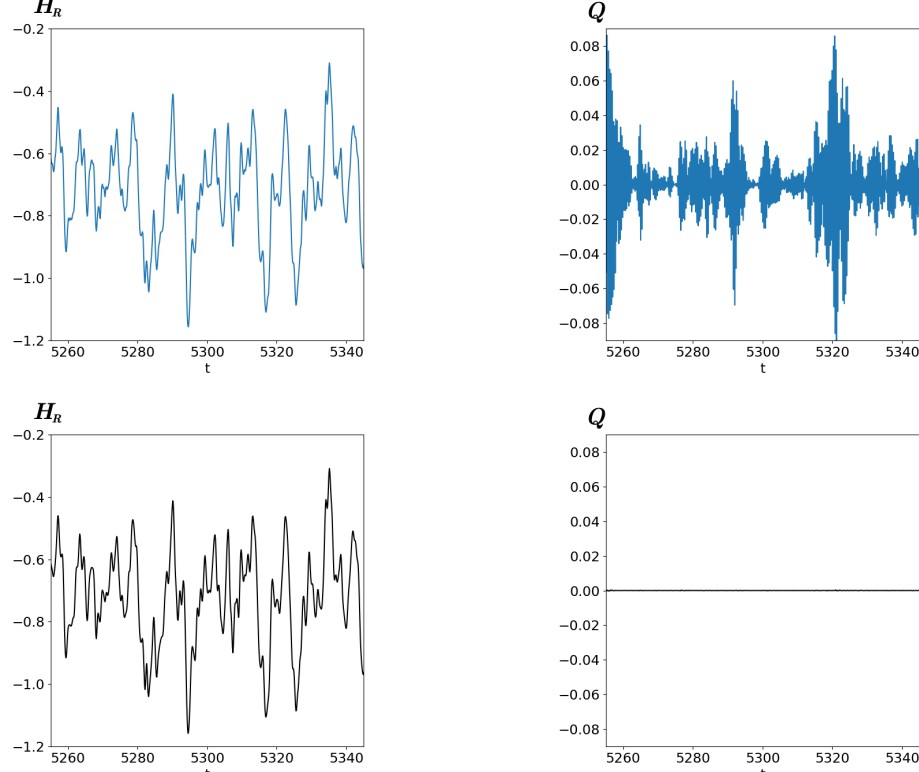

Figure 2: Long-time evolution of the different quantities considered. In *blue* are the plots corresponding to the *cat state* (protocol I) while in **black** are the plots for the *mixed state* **(protocol II)**. We see no qualitative difference for $H_R$ while the fluctuations of $Q$ around the mean are suppressed for the second protocol, as the consequence of the absence of initial quantum coherences. The blue-shaded region in the top-right panel is the consequence of oscillations occurring on a much shorter time scale.

meaning that the two protocols can be distinguished by looking at the fluctuations of $\hat{Q}$. Qualitatively, this comes from the fact that the observable $\hat{Q}$ has non zero projection on off-diagonal elements of the energy basis of $\hat{H}$. The fluctuations of the latter is precisely what characterizes the difference between the equilibrium state of protocol I and II. The computations and detailed expressions of these quantities are provided in app.C.

We show in fig.2, the time-evolution of $H_R(t)$ and $Q(t)$ in both protocols for a given realization of the disorder. We can clearly see that $H_R(t)$ is independent of the protocol, contrarily to $Q(t)$. The predicted value for the different quantities is also in quantitative arguments with the simulations (see tab.1). Details on how these values and the confidence intervals are obtained are given in app.C.

Let us add a remark here. In general because of its high degree of non-locality, it is not expected that $Q$ might be a suitable observable for experimental measurements. But similar qualitative statements about fluctuations should apply for any observables that couple the different energy sectors. For instance, as suggested at the end of [25], one could imagine doing an interference experiment between two parts of the system far part and look at the fluctuations of the pattern.

Table 1: Comparison between theoretical and numerical values for the mean and standard deviation of different quantities of interest.

| Theory | $\mathbb{E}[H_R]$ | $\sigma_{H_R}$ | $\sigma_Q$ |
|---|---|---|---|
| Cat state | $-7.13 * 10^{-1}$ | $1.51 * 10^{-1}$ | $2.09 * 10^{-2}$ |
| Mixed state | $-7.14 * 10^{-1}$ | $1.51 * 10^{-1}$ | $\approx 0$ |

| Numeric | $\mathbb{E}[H_R]$ | $\sigma_{H_R}$ | $\sigma_Q$ |
|---|---|---|---|
| Cat state | $-7.14 * 10^{-1} \pm 7.8 * 10^{-4}$ | $1.52 * 10^{-1} \pm 0.3 * 10^{-2}$ | $2.11 * 10^{-2} \pm 0.2 * 10^{-2}$ |
| Mixed state | $-7.14 * 10^{-1} \pm 7.8 * 10^{-4}$ | $1.52 * 10^{-1} \pm 0.3 * 10^{-2}$ | $5.83 * 10^{-5} \pm 1.0 * 10^{-6}$ |

## 4  Discussion

So far, we were only concerned about the *equilibrium state* of the system and haven't gone into the *thermalization* properties. Thermalization is stronger as it implies that the steady-state properties of the system can be described by one of the canonical ensemble of thermodynamics. In this section, we informally discuss possible links between the theory presented and the Eigenstate Thermalization Hypothesis (ETH). The ETH conjectures that for any initial state prepared as mixture of eigenstates of the total Hamiltonian with energies in a narrow window $[E - \delta E, E + \delta E]$, the matrix elements of observables in the energy eigenbasis is given by $O_{mn} = O(\overline{E})\delta_{mn} + e^{-S(\overline{E})/2} f_O(\overline{E}, \omega) R_{mn}$ with $\overline{E} \equiv (E_m + E_n)/2$, $\omega \equiv E_n - E_m$ and $S(\overline{E})$ the entropy. It is assumed that $O(\overline{E})$ and $f_O(\overline{E}, \omega)$ are *smooth* function of their arguments and that $R_{mn}$ is a random variable with zero mean and unit variance.

In the ETH, the role of the initial state is restricted to fixing the energy scales $E$ and $\Delta E$. The important remark is that, for the cat states, there is no notion of "narrow window" around a given energy anymore, hence we don't expect the ETH to apply. One illustration of that is the fact that off-diagonal correlations are not exponentially suppressed for cat states.

However the great advantage of the ETH is to explain why one can forget about all microscopic details contained in the diagonal ensemble and instead work with the microcanonical ensemble $\rho_{\mathrm{m}} \equiv \frac{1}{d}\mathbb{I}$. It would be great to have an equivalent statement here. A possible way for obtaining such simplification in our case already discussed in [26] would be the following : We can suppose that the "actual" group whose action leaves invariant the stationary state is not given by the set of all unitaries that commutes with $H$ but rather with an Hamiltonian $H' = H + \delta H$ with $\delta H$ a small perturbation which "mixes" the different energy sectors separated by energy $\approx \delta H$. The microcanonical ensemble is recovered in the case where the spectrum of $H'$ is fully degenerate in the energy window of interest $[E - \delta E, E + \delta E]$. Indeed, in that case, from (4) we see that the average density matrix is the microcanonical one : $\mathbb{E}[\rho_0] = \rho_{\mathrm{m}}$ and the second moment is $\mathbb{E}[\rho_0 \otimes \rho_0] = \frac{2}{d(d+1)}\mathbb{I}_{\{2\}}$ (for simplicity, we suppose that the initial state is a pure state). For the case where the initial state has two peaks $E_1$ and $E_2$ in its energy spectrum as shown in fig.1, we can conjecture that $H'$ is such that it mixes energies in the interval $I_1 = [E_1 - \delta E, E_1 + \delta E]$ and $I_2 = [E_2 - \delta E, E_2 + \delta E]$ around $E_1$ and $E_2$ but not altogether so that the average density matrix is given by : $\mathbb{E}[\rho_0] = \frac{\mathrm{tr}(\rho_0 \mathbb{I}_1)}{d_1}\mathbb{I}^{(1)} + \frac{\mathrm{tr}(\rho_0 \mathbb{I}_2)}{d_2}\mathbb{I}^{(2)}$ with $\mathbb{I}^{(i)} = \sum_{E \in I_i} |E\rangle \langle E|$ and $d_i = \mathrm{tr}(\mathbb{I}^{(i)})$. This has the simple interpretation that, on average at equilibrium, the mean density matrix is a statistical mixture of a state at energy $E_1$ and a state at energy $E_2$. The only information retained from the initial state is the weights corresponding to each energy sector. Similarly, a direct application of (6) shows that at the second order,

the information about the connected correlations between the two energy sectors is contained in a compact way in $\mathrm{tr}(\rho_0 \otimes \rho_0 \quad \mathbb{I}^{(1,2,2,1)})$ with $\mathbb{I}^{(1,2,2,1)} = \sum_{E \in I_1, E' \in I_2} |E, E'\rangle \langle E', E|$. Thus, one would not need fine information about the initial state to describe the equilibrium properties of the system. Of course, as with ETH, the range of applicability of these hypothesis is for now rather elusive and needs to be determined via careful numerical or experimental studies. We wish to report more on that in future studies.

## 5   Conclusion

We presented a theoretical framework enabling one to compute equilibrium properties of isolated quantum systems upon an assumption of *quantum ergodicity* which postulates that time-averages are equivalent to unitary ensemble averages in the stationary state. We brought specific attention to the relaxation of *cat states,* i.e quantum states which are a superposition of two macroscopically distinct states. We showed both analytically and numerically that a remnant of the initial quantum coherence was visible in the fluctuations around average quantities in the steady state whose amplitudes can be computed exactly. In the last part, we sketched a possible framework describing the equilibrium fluctuations in terms of statistical ensembles that do not require full knowledge of the microscopic details of the initial state.

In non-integrable or integrable systems a subject of debate of the previous decades has been to determine which conserved quantities were relevant to describe the thermal ensembles determining the local equilibrium properties. The question is of particular relevance for integrable systems since they comprise in principle a macroscopic number of conserved quantities [27]. It is now commonly accepted that one should only consider local (or quasilocal [28]) quantities to characterize such ensembles. However, our study stipulates that these ensembles no longer suffice when one looks at the equilibrium fluctuations of the system. There, additional information about possibly *non-local* conserved quantities are required.

Another important affirmation of ETH concerns the notion of *typicality* [29,30]. Typicality states that for all pure states that are random superpositions of eigenstates of the energy window, few-body operators have thermal distributions in the thermodynamic limit. It would be very interesting to test whether some notion of typicality remains in our case, i.e that the fluctuations of -possibly *non-local*- few-body observables are described by typical distributions in the thermodynamic limit starting from *any* superposition of states belonging to the macroscopically different energy sectors.

Another interesting point is that the equilibrium formulae (4,6) can in principle be applied to non chaotic or integrable system. Of course, there is no reason for the ergodic property to be fulfilled anymore so there is no guarantee that they provide the right predictions. For instance, for finite-size integrable systems, there might be long-lived oscillations that prevents the system from equilibrating [31,32]. However one should remark that the various symmetries of the system that lead to ergodicity breaking *are* accounted for in the structure of the group. It would therefore be interesting to see whether this information about degeneracies is enough to predict quantitatively the time-averaged and amplitudes of oscillating quantities and, if not, what ingredient needs to be added.

## Acknowledgements

This work wouldn't have been possible without past and present collaborations and discussions with D. Bernard and M. Bauer. I greatly benefited from crucial feedback from B. Appfel, P. Caucal, D. Martin and M. Rieu. I am also grateful for the work done by A. Gallin and T.

Orlovic. The numerics were performed using the QuSpin and QutiP Python packages.

**Funding information** I acknowledge support from the French École doctorale 564 and the Swiss National Science Foundation, Division II.

# A Second order fluctuations

In this section we show how to compute formula (6) from the main text , i.e we want to compute :

$$\mathbb{E}[\rho_0^{\otimes 2}] = \int_{\mathcal{G}} d\eta(U) U^{\otimes 2} \rho_0^{\otimes 2} U^{\dagger \otimes 2}.$$

In essence, the calculations will rely on the same mechanics than for order 1 with some twists. Once again, we define the decomposition of $\rho_0^{\otimes 2}$ into different sectors $(\rho_0^{\otimes 2})^{(i_1,j_1,i_2,j_2)}$ as follows :

$$(\rho_0^{\otimes 2})^{(i_1,j_1,i_2,j_2)} = \sum_{\nu_{i_1},\nu_{i_2},\nu_{j_1},\nu_{j_2}} \text{tr}(\rho_0^{\otimes 2} |E_{i_1}^{\nu_{i_1}}, E_{i_2}^{\nu_{i_2}}\rangle \langle E_{j_1}^{\nu_{j_1}}, E_{j_2}^{\nu_{j_2}}|) |E_{i_1}^{\nu_{i_1}}, E_{i_2}^{\nu_{i_2}}\rangle \langle E_{j_1}^{\nu_{j_1}}, E_{j_2}^{\nu_{j_2}}|.$$

The average of a block is given by :

$$\mathbb{E}[(\rho_0^{\otimes 2})^{(i_1,j_1,i_2,j_2)}] = \int_{\mathcal{G}} d\eta(U) U^{(i_1)} \otimes U^{(i_2)} (\rho_0^{\otimes 2})^{(i_1,j_1,i_2,j_2)} U^{\dagger (j_1)} \otimes U^{\dagger (j_2)}.$$

We first prove that $\mathbb{E}[(\rho_0^{\otimes 2})^{(i_1,j_1,i_2,j_2)}]$ is null except if the tuple $\{j_1, j_2\}$ is a permutation of $\{i_1, i_2\}$. Indeed, suppose it's not the case : then, there exists a $k$ such that $\forall k'$, $i_k \neq j_{k'}$. For definiteness, say $k = 1$. From the left invariance of the Haar measure, we then have $\forall V^{(i_1)} \in SU(d_{i_1})$ that :

$$V^{(i_1)} \otimes \mathbb{I}^{(i_2)} \mathbb{E}[(\rho_0^{\otimes 2})^{(i_1,j_1,i_2,j_2)}] = \mathbb{E}[(\rho_0^{\otimes 2})^{(i_1,j_1,i_2,j_2)}]$$
$$(V^{(i_1)} \otimes \mathbb{I}^{(i_2)} - \mathbb{I}^{(i_1)} \otimes \mathbb{I}^{(i_2)}) \mathbb{E}[(\rho_0^{\otimes 2})^{(i_1,j_1,i_2,j_2)}] = 0,$$

which implies $\mathbb{E}[(\rho_0^{\otimes 2})^{(i_1,j_1,i_2,j_2)}] = 0$.

We thus learn the important fact that $\{j_1, j_2\}$ must be a permutation of $\{i_1, i_2\}$ for the average of the block to be non zero.

There are three possible cases that we will examine separately :

I $i_1 = i_2 = j_1 = j_2$,

II $i_1 \neq i_2$, $i_1 = j_1$, $i_2 = j_2$ corresponding to the permutation $\sigma \in \mathfrak{S}_2 : \{1,2\} \to \{1,2\}$,

III $i_1 \neq i_2$, $i_1 = j_2$, $i_2 = j_1$ corresponding to the permutation $\sigma \in \mathfrak{S}_2 : \{1,2\} \to \{2,1\}$.

**Case I :**

$$[(\rho_0^{\otimes 2})^{(i_1 i_1 i_1 i_1)}] = \int d\eta(U) U^{(i_1)} \otimes U^{(i_1)} (\rho_0 \otimes \rho_0)^{(i_1 i_1 i_1 i_1)} U^{\dagger (i_1)} \otimes U^{\dagger (i_1)}.$$

Let $D^{(i)}$ be the fundamental representation of $SU(d_i)$. The tensor product representation $D^{(i)} \otimes D^{(i)}$ admits a decomposition onto irreducible representations indexed by Young diagrams. We denote by $\{2\}$ the possible partitions of 2, i.e $(2)$ and $(1,1)$. Following the usual convention for indexation of irreducible representations of the unitary group by Young tableaux, $D^{(2)(i)}$ corresponds to ▢▢ and the tensor representation that is symmetric under permutation of two

indices while $D^{(1,1)(i)}$ corresponds to ▢ and denotes the antisymmetric representation. We have [33] :

$$D^{(i)} \otimes D^{(i)} = D^{(2)(i)} \oplus D^{(1,1)(i)}.$$

The representation $D^{(2)(i)}$ preserves the symmetric eigenbasis made of $\frac{d_i(d_i+1)}{2}$ elements $\left|E_i,(2),v_i,v_i'\right\rangle \equiv \frac{1}{\sqrt{2}}(|E_i^{v_i},E_i^{v_i'}\rangle + |E_i^{v_i'},E_i^{v_i}\rangle)$ for $v_i \neq v_i'$ and $|E_i,(2),v_i,v_i\rangle = \left|E_i^{v_i},E_i^{v_i}\right\rangle$ while the representation $D^{(1,1)(i)}$ preserves the antisymmetric eigenbasis made of $\frac{d_i(d_i-1)}{2}$ elements $\left|E_i,(1,1),v_i,v_i'\right\rangle = \frac{1}{\sqrt{2}}(|E_i^{v_i},E_i^{v_i'}\rangle - |E_i^{v_i'},E_i^{v_i}\rangle)$ for $v_i \neq v_i'$.

One can further block-decompose $(\rho_0 \otimes \rho_0)^{(i_1,i_1,i_1,i_1)}$ according to these basis. Introducing :

$$
\begin{aligned}
&(\rho_0^{\otimes 2})^{(i_1,i_1,i_1,i_1),(y_1,y_2)} \\
&\equiv \sum_{v_i,v_i',\mu_i,\mu_i'} \text{tr}((\rho_0^{\otimes 2})^{(i_1,i_1,i_1,i_1)} \left|E_{i_1},(y_1),v_{i_1},v_{i_1}'\right\rangle \left\langle E_{i_1},(y_2),\mu_{i_1},\mu_{i_1}'\right|) \\
&\qquad \left|E_{i_1},(y_1),v_{i_1},v_{i_1}'\right\rangle \left\langle E_{i_1},(y_2),\mu_{i_1},\mu_{i_1}'\right|,
\end{aligned}
$$

with $y_1, y_2 \in \{2\}$.

By Schur lemma, we then have than the only non-zero block components of $\mathbb{E}[(\rho_0^{\otimes 2})^{(i_1 i_1 i_1 i_1)}]$ are the diagonal ones, i.e $\mathbb{E}[(\rho_0^{\otimes 2})^{(i_1 i_1 i_1 i_1),((2),(2))}]$ and $\mathbb{E}[(\rho_0^{\otimes 2})^{(i_1 i_1 i_1 i_1),((1,1),(1,1))}]$ and these blocks are proportional to the identity :

$$
\begin{aligned}
\mathbb{E}[(\rho_0^{\otimes 2})^{(i_1 i_1 i_1 i_1),((2),(2))}] &\propto \mathbb{I}_{(2)}^{(i_1,i_1,i_1,i_1)}, \\
\mathbb{E}[(\rho_0^{\otimes 2})^{(i_1 i_1 i_1 i_1),((1,1),(1,1))}] &\propto \mathbb{I}_{(1,1)}^{(i_1,i_1,i_1,i_1)}.
\end{aligned}
$$

Where $\mathbb{I}_{\{2\}}^{(i_1,i_1,i_1,i_1)}$ is the identity matrix associated to the representation $D^{\{2\}(i_1)}$. The proportionality coefficient is determined by taking the trace. Explicitly, we have :

$$
\begin{aligned}
\mathbb{E}[(\rho_0^{\otimes 2})^{(i_1 i_1 i_1 i_1),((2),(2))}] &= \frac{2}{d_{i_1}(d_{i_1}+1)} \text{tr}((\rho_0^{\otimes 2})\mathbb{I}_{(2)}^{(i_1,i_1,i_1,i_1)})\mathbb{I}_{(2)}^{(i_1,i_1,i_1,i_1)} \\
&= \frac{1}{d_{i_1}(d_{i_1}+1)}((\text{tr}(\rho_0^{(i_1)}))^2 + \text{tr}((\rho_0^{(i_1)})^2))\mathbb{I}_{(2)}^{(i_1,i_1,i_1,i_1)} \\
\mathbb{E}[(\rho_0^{\otimes 2})^{(i_1 i_1 i_1 i_1),((1,1),(1,1))}] &= \frac{2}{d_i(d_i-1)} \text{tr}((\rho_0^{\otimes 2})\mathbb{I}_{(1,1)}^{(i_1,i_1,i_1,i_1)})\mathbb{I}_{(1,1)}^{(i_1,i_1,i_1,i_1)}. \\
&= \frac{1}{d_i(d_i-1)}((\text{tr}(\rho_0^{(i_1)}))^2 - \text{tr}((\rho_0^{(i_1)})^2))\mathbb{I}_{(1,1)}^{(i_1,i_1,i_1,i_1)}
\end{aligned}
$$

**Case II :** We look at :

$$\mathbb{E}[(\rho_0^{\otimes 2})^{(i_1 i_2 i_1 i_2)}] = \int d\eta(U)U^{(i_1)} \otimes U^{(i_2)}(\rho_0^{\otimes 2})^{(i_1 i_2 i_1 i_2)}U^{\dagger(i_1)} \otimes U^{\dagger(i_2)}$$

for $i_1 \neq i_2$.

Since $D^{(i_1)}$ and $D^{(i_2)}$ are two irreducible representations of $SU(d_{i_1})$ and $SU(d_{i_2})$, the tensor product $D^{(i_1)} \otimes D^{(i_2)}$ is an irreducible representation of $SU(d_{i_1}) \times SU(d_{i_2})$.

The proof comes from *Schur orthogonality relation* :

Let $D_1$ and $D_2$ be two irreducible representations over vector spaces $V_1$ and $V_2$ : $D_1 : G1 \to \text{End}(V1)$, $D_2 : G2 \to \text{End}(V2)$.

If $G1$ and $G2$ are finite, we show that the tensor product representation $D_\otimes = D_1 \otimes D_2 : G1 \times G2 \to \text{End}(V1 \otimes V2)$, defined for any couple $(g1, g2) \in G1 \times G2$ by $D_\otimes(g1, g2) = D1(g1) \otimes D2(g2)$

is again irreducible. Indeed the Schur orthogonality relation for an irreducible representation states (with normalized measure with respect to the group volume) that :

$$\int d\eta(g)|\chi(g)|^2 = 1,$$

where $\chi(g)$ is the character of the representation. Then :

$$
\begin{aligned}
\int d\eta(g_1 \times g_2)|\chi_\otimes(g_1 \times g_2)|^2 &= \int d\eta(g_1)|\chi_1(g_1)|^2 \int d\eta(g_2)|\chi_2(g_2)|^2 \\
&= 1
\end{aligned}
$$

and the representation $D_\otimes$ is again irreducible.

By Schur lemma, we then have that $\mathbb{E}[(\rho_0^{\otimes 2})^{(i_1 i_2 i_1 i_2)}] \propto \mathbb{I}^{(i_1 i_2 i_1 i_2)}$ where $\mathbb{I}^{(i_1 i_2 i_1 i_2)}$ is the identity defined by : $\mathbb{I}^{(i,j,k,l)} \equiv \sum_{\nu_i, \nu_j, \nu_k, \nu_l} |E_i^{\nu_i}, E_j^{\nu_j}\rangle \langle E_k^{\nu_k}, E_l^{\nu_l}|$. As before, we take the trace to determine the proportionality coefficient, we get :

$$\mathbb{E}[(\rho_0^{\otimes 2})^{(i_1 i_2 i_1 i_2)}] = \frac{\operatorname{tr}(\rho_0^{(i_1 i_1)})\operatorname{tr}(\rho_0^{(i_2 i_2)})}{d_{i_1} d_{i_2}}\mathbb{I}^{(i_1 i_2 i_1 i_2)}.$$

**Case III :** We look at :

$$\mathbb{E}[(\rho_0^{\otimes 2})^{(i_1 i_2 i_2 i_1)}] = \int d\eta(U) U^{(i_1)} \otimes U^{(i_2)} (\rho_0^{\otimes 2})^{(i_1 i_2 i_2 i_1)} U^{\dagger(i_2)} \otimes U^{\dagger(i_1)}.$$

Let $M$ be an element of $M \in L(\mathcal{H}_i, \mathcal{H}_j) \otimes L(\mathcal{H}_k, \mathcal{H}_l)$ and $\sigma \in \mathfrak{S}_2$. We define the *right action* of $\sigma$ on $M$, $M \cdot \sigma$ as :

$$(M \cdot \sigma)_{\alpha_1 \beta_1 \alpha_2 \beta_2} = M_{\alpha_1 \beta_{\sigma(1)} \alpha_2 \beta_{\sigma(2)}}.$$

As it will be useful later, we define in the same way, the *left action* $\sigma \cdot M$ acting on $M$ as :

$$(\sigma \cdot M)_{\alpha_1 \beta_1 \alpha_2 \beta_2} = M_{\alpha_{\sigma(1)} \beta_1 \alpha_{\sigma(2)} \beta_2}.$$

We then have that :

$$
\begin{aligned}
\mathbb{E}[(\rho_0^{\otimes 2})^{(i_1 i_2 i_2 i_1)}] &= \int d\eta(U) U^{(i_1)} \otimes U^{(i_2)} (\rho_0^{\otimes 2})^{(i_1 i_2 i_2 i_1)} U^{\dagger(i_{\sigma(1)})} \otimes U^{\dagger(i_{\sigma(2)})} \\
\mathbb{E}[(\rho_0^{\otimes 2})^{(i_1 i_2 i_2 i_1)}] \cdot \sigma &= \left(\int d\eta(U) U^{(i_1)} \otimes U^{(i_2)} ((\rho_0^{\otimes 2})^{(i_1 i_2 i_2 i_1)} . \sigma) U^{\dagger(i_1)} \otimes U^{\dagger(i_2)}\right),
\end{aligned}
$$

where $\sigma$ is here the permutation $\{1, 2\} \to \{2, 1\}$.

Applying Schur lemma as before then leads to :

$$
\begin{aligned}
\mathbb{E}[(\rho_0^{\otimes 2})^{(i_1 i_2 i_2 i_1)}] \cdot \sigma &= \frac{\operatorname{tr}((\rho_0^{\otimes 2})^{(i_1 i_2 i_2 i_1)} . \sigma)}{d_{i_1} d_{i_2}}\mathbb{I}^{(i_1 i_1 i_2 i_2)}, \\
\mathbb{E}[(\rho_0^{\otimes 2})^{(i_1 i_2 i_2 i_1)}] &= \frac{\operatorname{tr}(\rho_0^{(i_1 i_2)} \rho_0^{(i_2 i_1)})}{d_{i_1} d_{i_2}}\mathbb{I}^{(i_1 i_2 i_2 i_1)}, \\
&= \frac{\operatorname{tr}((\rho_0^{\otimes 2})\mathbb{I}^{(i_1 i_2 i_2 i_1)})}{d_{i_1} d_{i_2}}\mathbb{I}^{(i_1 i_2 i_2 i_1)}.
\end{aligned}
$$

Regrouping the results for all three cases proves (6) of the main text :

$$
\begin{aligned}
\mathbb{E}[\rho_0^{\otimes 2}] \;=\; & \sum_{i_1} \frac{1}{d_{i_1}(d_{i_1}+1)}\big((\mathrm{tr}(\rho_0^{(i_1)}))^2 + \mathrm{tr}((\rho_0^{(i_1)})^2)\big)\mathbb{I}_{(2)}^{(i_1,i_1,i_1,i_1)} \\
& + \frac{1}{d_{i_1}(d_{i_1}-1)}\big((\mathrm{tr}(\rho_0^{(i_1)}))^2 - \mathrm{tr}((\rho_0^{(i_1)})^2)\big)\mathbb{I}_{(1,1)}^{(i_1,i_1,i_1,i_1)} \\
& + \sum_{i_1\neq i_2} \frac{1}{d_{i_1}d_{i_2}}\big(\mathrm{tr}(\rho_0^{(i_1 i_1)})\mathrm{tr}(\rho_0^{(i_2 i_2)})\mathbb{I}^{(i_1 i_2 i_1 i_2)} + \mathrm{tr}(\rho_0^{(i_1 i_2)}\rho_0^{(i_2 i_1)})\mathbb{I}^{(i_1 i_2 i_2 i_1)}\big).
\end{aligned}
$$

## B  General formula at any order

In this section, we wish to compute the generalization of the formula for the mean and second order correlation of density matrix elements to higher order, i.e :

$$
\mathbb{E}[\rho_0^{\otimes n}] \;=\; \int_{\mathcal{G}} d\eta(U)\,\mathrm{Tr}(U^{\otimes n}\rho_0^{\otimes n}U^{\dagger\otimes n}). \tag{16}
$$

This is equivalent to knowing the generating function $Z(A)$ defined as

$$
Z(A) \equiv \int_{\mathcal{G}} d\eta(U)\, e^{\mathrm{tr}(AU\rho_0 U^{\dagger})},
$$

which is reminiscent of the Harish-Chandra Itzykson Zuber integral [34] except that the group $\mathcal{G}$ upon which the integration is performed is *not* an unitary group so we can't directly use that result.

   To compute (16), we will rely on the same approach than for the mean and the second order correlations, i.e we will first decompose $\rho_0^{\otimes n}$ into different *sectors* transforming according to different combination of $U^{(i_k)}$ and identify the different *invariants* under such transformations. In spirit, it will be close to the proof of the invariant theory presented in the appendix of [11]. We introduce once again the block decomposition of $\rho_0^{\otimes n}$ into $(\rho_0^{\otimes n})^{(ij)} \equiv (\rho_0^{\otimes n})^{(i_1 i_2 \cdots i_n, j_1 j_2 \cdots j_n)}$ defined by :

$$
\begin{aligned}
& (\rho_0^{\otimes n})^{(i_1 i_2 \cdots i_n, j_1 j_2 \cdots j_n)} \\
& = \sum_{\nu_{i_1},\cdots,\nu_{i_n},\nu_{j_1},\cdots,\nu_n} \mathrm{tr}(\rho_0^{\otimes n}|E_{i_1}^{\nu_{i_1}},\cdots,E_{i_n}^{\nu_{i_n}}\rangle\langle E_{j_1}^{\nu_{j_1}},\cdots,E_{j_n}^{\nu_{j_n}}|)|E_{i_1}^{\nu_{i_1}},\cdots,E_{i_n}^{\nu_{i_n}}\rangle\langle E_{j_1}^{\nu_{j_1}},\cdots,E_{j_n}^{\nu_{j_n}}|.
\end{aligned}
$$

The average of a block is given by :

$$
\mathbb{E}[(\rho_0^{\otimes n})^{(ij)}] = \int_{U\in\mathcal{G}} d\eta(U)\, U^{(i_1)}\otimes U^{(i_2)}\cdots\otimes U^{(i_n)}(\rho_0^{\otimes n})^{(ij)}U^{\dagger(j_1)}\otimes U^{\dagger(j_2)}\cdots\otimes U^{\dagger(j_n)}.
$$

To each term $U^{(i_1)}\otimes U^{(i_2)}\cdots\otimes U^{(i_n)}$, we associate a *standard ordering* defined as the tensor product

$$
\begin{aligned}
U_{\mathrm{so}}^{(m_1\cdots m_{k_{\max}})} \;\equiv\; & U^{(i_{\sigma^{-1}(1)})}\otimes U^{(i_{\sigma^{-1}(2)})}\cdots\otimes U^{(i_{\sigma^{-1}(n)})} \\
\equiv\; & U^{(m_1)\otimes n_1}\otimes\cdots\otimes U^{(m_n)\otimes n_k},
\end{aligned}
$$

where $i_{\sigma^{-1}(1)} \le i_{\sigma^{-1}(2)}\cdots \le i_{\sigma^{-1}(n)}$, $\sigma\in\frac{\mathfrak{S}_n}{\prod\mathfrak{S}_{n_k}}$, $m_k = i_{\sigma^{-1}(\sum_{j=1}^k n_j)}$ and $n_k$ is the number of times $U^{(m_k)}$ appears in the tensor product (we have $\sum_{k=1}^{k_{\max}} n_k = n$).

In the same way, we define the permutation $\sigma^* \in \frac{\mathfrak{S}_n}{\prod \mathfrak{S}_{n_k}}$ such that $U^{\dagger(j_{\sigma^{*-1}(1)})} \otimes U^{\dagger(j_{\sigma^{*-1}(2)})} \cdots \otimes U^{\dagger(j_{\sigma^{*-1}(n)})}$ is standard ordered.

We can use the same argument as before to show that the average is null unless the $j'_k s$ are a permutation of the $i_k$'s. We call this permutation $\gamma : j_{\gamma(k)} = i_k$. $\gamma$ is related to $\sigma, \sigma^*$ by $\gamma = \sigma^{*-1}\sigma$. Indeed :

$$
\begin{aligned}
i_{\sigma^{-1}(k)} &= j_{\sigma^{*-1}(k)} \\
i_k &= j_{\sigma^{*-1}(\sigma(k))} \\
j_{\gamma(k)} &= j_{\sigma^{*-1}(\sigma(k))}.
\end{aligned}
$$

Now :

$$
\mathbb{E}[(\rho_0^{\otimes n})^{(ij)}] = \int_{U \in \mathcal{G}} d\eta(U) U^{(i_{\sigma(\sigma^{-1}(1))})} \cdots \otimes U^{(i_{\sigma(\sigma^{-1}(n))})} (\rho_0^{\otimes n})^{(ij)} U^{(j_{\sigma^*(\sigma^{*-1}(1))})} \cdots \otimes U^{\dagger(j_{\sigma^*(\sigma^{*-1}(n))})}
$$

$$
\sigma \cdot \mathbb{E}[(\rho_0^{\otimes n})^{(ij)}] \cdot \sigma^* = \int_{U \in \mathcal{G}} d\eta(U) U_{\mathrm{so}}^{(m_1, \cdots, m_{k_{\max}})} \sigma \cdot (\rho_0^{\otimes n})^{(ij)} \cdot \sigma^* U_{\mathrm{so}}^{\dagger(m_1, \cdots, m_{k_{\max}})}.
$$

In general, the tensor product representation which $U_{\mathrm{so}}^{(m_1, \cdots, m_{k_{\max}})}$ belongs to is reducible :

$$
\begin{aligned}
U_{\mathrm{so}}^{(m_1, \cdots, m_{k_{\max}})} &= U^{(m_1) \otimes n_1} \otimes \cdots \otimes U^{(m_{k_{\max}}) \otimes n_{k_{\max}}}. \\
&= \oplus_{(y_1) \in \{n_1\}} D^{(y_1)}(U^{(m_1)}) \otimes \oplus_{(y_2) \in \{n_2\}} D^{(y_2)}(U^{(m_2)}) \cdots \otimes \oplus_{y_{k_{\max}} \{n_{k_{\max}}\}} D^{(y_{k_{\max}})}(U^{(m_{k_{\max}})}) \\
&= \oplus_{\{n_1\}, \{n_2\}, \cdots \{n_{k_{\max}}\}} D^{(y_1)}(U^{(m_1)}) \otimes \cdots \otimes D^{(y_{k_{\max}})}(U^{(m_{k_{\max}})}) \\
&\equiv \oplus_{\{n\}} \otimes_i D^{(y_i)}(U^{(m_i)}),
\end{aligned}
$$

where as before $\{n_k\}$ designates the possible Young tableaux of $n_k$. As we showed before, the tensor product of two irreducible representations is again irreducible, so the representations $D^{(y_1)} \otimes \cdots \otimes D^{(y_{k_{\max}})}$ of $\times_i SU(d_i)^{n_k}$ are *irreducible*. As before, we decompose further $\sigma \cdot (\rho_0^{\otimes n})^{(ij)}.\sigma^*$ into blocks corresponding to these irreducible representations. Denoting by $|E_i, (y), \nu\rangle$ the basis elements associated to the irreducible representation of $SU(d_i)^n$ corresponding to the decomposition $(y)$, we have the following decomposition for the blocks :

$$
(\sigma.(\rho_0^{\otimes n})^{(i,j)}.\sigma^*)^{(\{y\}, \{y'\})}
$$

$$
= \sum_{\nu_1, \cdots, \nu_{k_{\max}}, \nu'_1, \cdots, \nu'_{k_{\max}}}
$$

$$
\mathrm{tr}((\sigma \cdot (\rho_0^{\otimes n})^{(ij)}.\sigma^*) \left| E_{m_1}, (y_1), \nu_1, \cdots, E_{m_{k_{\max}}}, (y_{k_{\max}}), \nu_{k_{\max}} \right\rangle \left\langle E_{m_1}, (y'_1), \nu'_1, \cdots, E_{m_{k_{\max}}}, (y'_{k_{\max}}), \nu'_{k_{\max}} \right|)
$$

$$
\left| E_{m_1}, (y_1), \nu_1, \cdots, E_{m_{k_{\max}}}, (y_{k_{\max}}), \nu_{k_{\max}} \right\rangle \left\langle E_{m_1}, (y'_1), \nu'_1, \cdots, E_{m_{k_{\max}}}, (y'_{k_{\max}}), \nu'_{k_{\max}} \right|,
$$

where $\{y\}$ designates the tuple $\{y_j\}_{1 \le j \le k_{\max}}$. The averages of these blocks are given by

$$
\begin{aligned}
&\mathbb{E}[(\sigma.(\rho_0^{\otimes n})^{(i,j)}.\sigma^*)^{(\{y\}, \{y'\})}] \\
&= \int_{\mathcal{G}} D^{(y_1)}(U^{(m_1)}) \otimes \cdots \otimes D^{(y_{k_{\max}})}(U^{(m_{k_{\max}})}) (\sigma.(\rho_0^{\otimes n})^{(i,j)}.\sigma^*)^{(\{y\}, \{y'\})} \\
&\quad D^{\dagger(y'_1)}(U^{(m'_1)}) \otimes \cdots \otimes D^{\dagger(y'_{k_{\max}})}(U^{(m'_{k_{\max}})}).
\end{aligned}
$$

As before, by Schur lemma, the average of one of these blocks is non zero only if the two representations are equivalent, meaning that we must have $\{y\} = \{y'\}$. Then :

$$
\mathbb{E}[(\sigma.(\rho_0^{\otimes n})^{(i,j)}.\sigma^*)^{(\{y\}, \{y'\})}] = \delta_{y, y'} \prod_k \frac{\omega((y_k))}{d_{(y_k)}} \mathrm{tr}((\sigma \cdot (\rho_0^{\otimes n})^{(ij)} \cdot \sigma^*)^{(\{y\}, \{y\})}) (\sigma.\mathbb{I}^{(i,j)}.\sigma^*)^{(\{y\}, \{y\})},
$$

where $\omega((y_k))$ is the multiplicity of the Young tableau corresponding to $(y_k)$, $d_{(y_k)}$ its dimension and $(\sigma.\mathbb{I}^{(ij)}.\sigma^*)^{(\{y\}\{y\})}$ is defined as :

$$
\begin{aligned}
&(\sigma.\mathbb{I}^{(ij)}.\sigma^*)^{(\{y\}\{y\})} \\
&\equiv \sum_{v_1,\cdots v_{k_{\max}}} \Big| E_{m_1},(y_1),v_1,\cdots,E_{m_{k_{\max}}},(y_{k_{\max}}),v_{k_{\max}} \Big\rangle \Big\langle E_{m_1},(y_1),v_1,\cdots,E_{m_{k_{\max}}},(y_{k_{\max}}),v_{k_{\max}} \Big|.
\end{aligned}
$$

This finally leads us to :

$$
\mathbb{E}[\rho_0^{\otimes n}] = \sum_{(i,j)}\sum_{(\{y\})}\prod_k \frac{\omega((y_k))}{d_{(y_k)}} \mathrm{tr}((\sigma \cdot (\rho_0^{\otimes n})^{(ij)} \cdot \sigma^*)^{(\{y\},\{y\})})\mathbb{I}^{(ij)(\{y\},\{y\})},
$$

where the sum is over all $j$'s that are a permutation of $i$'s. Each permutation is characterized by $\sigma$ and $\sigma^*$.

## C   More details on the case study

In this appendix, we provide additional details on the numerics presented in the main text. As a reminder, the Hamiltonian we study is :

$$
\hat{H} = \sum_{j=0}^{L-2} J\vec{\sigma}_j \cdot \vec{\sigma}_{j+1} + \sum_{j=0}^{L-1} h_j \sigma_j^z,
$$

with $L = 12$, $J = 1$ and $h_j$ picked at random between $-1$ and $1$ with the uniform distribution. We work in the 0 magnetization sector which has a dimension of 924. We choose a seed such that the mean level spacing of the spectrum is close to the Wigner distribution one : 0.53069. The minimum energy $E_{\min}$ is $-20.944$ and the maximum energy $E_{\max}$ is 12.445. The states $|\Phi_1\rangle$ and $|\Phi_2\rangle$ have respectively energies $E_1 \equiv -10.753$ and $E_2 \equiv 6.731$ with respect to $\hat{H}_R + \hat{H}_L$. One important quantity is the overlap $O_{\mathrm{vlap}}$ these states have with respect to the eigenbasis $|i\rangle$ of the total Hamiltonian, i.e $O_{\mathrm{vlap}}(i) \equiv |\langle i|\Phi_1\rangle \langle i|\Phi_2\rangle|$. The maximum of $O_{\mathrm{vlap}}$ in our case is $8.206 * 10^{-5}$. We also have that $\sum_i O_{\mathrm{vlap}}(i) = 0.00579$. The decompositions of $|\Phi_1\rangle$ and $|\Phi_2\rangle$ in the eigenbasis of $\hat{H}$ are shown on fig.3.

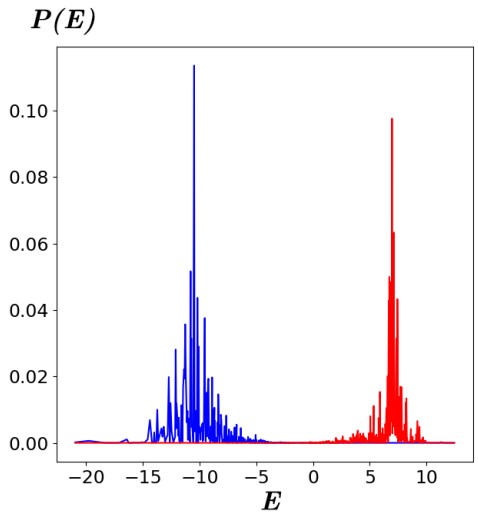

Figure 3: Decomposition of $|\Phi_1\rangle$ (blue) and $|\Phi_2\rangle$ (red) in the eigenbasis of $\hat{H}$.

To compute the numerical values, we choose a time window $[3000 : 13000]$ with $2 * 10^4$ points. The different values presented in tab.1 of the main text are given by time average over this interval :

$$\mathbb{E}_t[A(t)] = \frac{1}{T}\int_0^T dt\, A(t)$$

$$\sigma_A \equiv \sqrt{\frac{1}{T}\int_0^T (A(t) - \mathbb{E}_t[A(t)])^2},$$

with $T$ the time interval.

The confidence intervals $\delta$ on the mean and the standard deviation are obtained by dividing $T$ in 10 smaller intervals on which the quantities of interest are computed again. $\delta$ then corresponds to the standard deviation between the results obtained on the 10 samples and the one computed on the full interval.

The theoretical values for the mean and variances of our different quantities are computed from equations (4) and (6) of the main text. Recall that for the protocol I, the initial state was chosen as the pure state $\frac{1}{\sqrt{2}}(|\Phi_1\rangle + |\Phi_2\rangle)$ and for the protocol II, it was the classical mixture described by the density matrix $\rho_0 = \frac{1}{2}(|\Phi_1\rangle\langle\Phi_1| + |\Phi_2\rangle\langle\Phi_2|)$.

For a fully degenerate spectrum, (4,6) gives that the average of a given observable $\hat{M}$ and its second order connected correlations are given by :

$$\text{tr}(\mathbb{E}[\rho_0]\hat{M}) = \sum_i \langle i\,|\rho_0|\,i\rangle\langle i|\,\hat{M}\,|i\rangle$$

$$\text{tr}(\mathbb{E}[\rho_0^{\otimes 2}]\hat{M}\otimes\hat{M})^c = \sum_{i_1\neq i_2} |\langle i_1\,|\rho_0|\,i_2\rangle|^2 |\langle i_1|\,\hat{M}\,|i_2\rangle|^2.$$

Below, we explain why there is no difference in the first and second order correlation of $\hat{H}_R$ between protocol I and II while there are for the second order correlation of $\hat{Q}$.

**Mean quantities :**

$$\text{tr}(\mathbb{E}[\rho_0^{(I)}]\hat{H}_R) = \sum_i \langle i\,|\rho_0|\,i\rangle\langle i|\,\hat{H}_R\,|i\rangle$$

$$= \frac{1}{2}\sum_i \langle i|\,(|\Phi_1\rangle + |\Phi_2\rangle)(\langle\Phi_1| + \langle\Phi_2|)\,|i\rangle\langle i|\,\hat{H}_R\,|i\rangle. \tag{17}$$

Since the overlap between $|\Phi_1\rangle$ and $|\Phi_2\rangle$ is small, we can approximate the previous expression by :

$$\text{tr}(\mathbb{E}[\rho_0^{(I)}]\hat{H}_R) \approx \frac{1}{2}\sum_i \langle i|\,\hat{H}_R\,|i\rangle\langle i|\,(|\Phi_1\rangle\langle\Phi_1| + |\Phi_2\rangle\langle\Phi_2|)\,|i\rangle$$

$$= \text{tr}(\mathbb{E}[\rho_0^{(II)}]\hat{H}_R).$$

Similarly :

$$\text{tr}(\mathbb{E}[\rho_0^{(I)}]\hat{Q}) = \sum_i \langle i|\Phi_1\rangle\langle\Phi_2|i\rangle\langle i|\,\rho_0^{(I)}\,|i\rangle$$

$$\approx 0$$

$$= \text{tr}(\mathbb{E}[\rho_0^{(II)}]\hat{Q}).$$

We see that as far as the mean quantities are concerned, the cat state and the classical mixture provide the same results.

**Second order fluctuations :**    From (6), we have that :

$$\mathrm{tr}(\mathbb{E}[\rho_0^{(\mathrm{I})\otimes 2}]\hat{H}_\mathrm{R}\otimes\hat{H}_\mathrm{R})^\mathrm{c}$$

$$= \sum_{i_1\neq i_2} |\langle i_1|\rho_0^{(\mathrm{I})}|i_2\rangle|^2 \langle i_1|\hat{H}_\mathrm{R}|i_2\rangle|^2$$

$$= \frac{1}{4}\sum_{i_1\neq i_2} |\langle i_1|(|\Phi_1\rangle\langle\Phi_1|+|\Phi_1\rangle\langle\Phi_2|+|\Phi_2\rangle\langle\Phi_1|+|\Phi_2\rangle\langle\Phi_2|)|i_2\rangle|^2|\langle i_1|\hat{H}_\mathrm{R}|i_2\rangle|^2$$

$$\approx \frac{1}{4}\sum_{i_1\neq i_2} |\langle i_1|(|\Phi_1\rangle\langle\Phi_1|+|\Phi_2\rangle\langle\Phi_2|)|i_2\rangle|^2|\langle i_1|\hat{H}_\mathrm{R}|i_2\rangle|^2$$

$$\approx \mathrm{tr}(\mathbb{E}[\rho_0^{(\mathrm{II})\otimes 2}]\hat{H}_\mathrm{R}\otimes\hat{H}_\mathrm{R})^\mathrm{c}.$$

Where in the last line we used the fact that $\langle i_1|\Phi_1\rangle\langle\Phi_2|i_2\rangle$ is non zero only for $E_{i_1}$ close to $E_1$ and $E_{i_2}$ close to $E_2$. But this in turns imply that $\langle i_1|\hat{H}_\mathrm{R}|i_2\rangle\approx 0$ .

For $\hat{Q}$ :

$$\mathrm{tr}(\mathbb{E}[\rho_0^{(\mathrm{I})\otimes 2}]\hat{Q}\otimes\hat{Q})^\mathrm{c}$$

$$= \frac{1}{4}\sum_{i_1\neq i_2} |\langle i_1|(|\Phi_1\rangle\langle\Phi_1|+|\Phi_1\rangle\langle\Phi_2|+|\Phi_2\rangle\langle\Phi_1|+|\Phi_2\rangle\langle\Phi_2|)|i_2\rangle|^2|\langle i_1|\hat{Q}|i_2\rangle|^2$$

$$= \frac{1}{4}\sum_{i_1\neq i_2} |\langle i_1|(|\Phi_1\rangle\langle\Phi_1|+|\Phi_1\rangle\langle\Phi_2|+|\Phi_2\rangle\langle\Phi_1|+|\Phi_2\rangle\langle\Phi_2|)|i_2\rangle|^2|\langle i_1|\Phi_1\rangle\langle\Phi_2|i_2\rangle+\langle i_1|\Phi_2\rangle\langle\Phi_1|i_2\rangle|^2$$

$$\approx \frac{1}{4}\sum_{i_1\neq i_2} |\langle i_1|\Phi_1\rangle\langle\Phi_2|i_2\rangle+\langle i_1|\Phi_2\rangle\langle\Phi_1|i_2\rangle|^4.$$

Where to go to the last line we used the fact that if $|\langle i_1|\Phi_1\rangle\langle\Phi_2|i_2\rangle+\langle i_1|\Phi_2\rangle\langle\Phi_1|i_2\rangle|^2$ is non zero, the $E_{i_1}$ is close to either $E_1$ or $E_2$ and $E_{i_2}$ the other way around. This in turn implies that $\langle i_1|\Phi_1\rangle\langle\Phi_1|i_2\rangle\approx 0\approx\langle i_1|\Phi_2\rangle\langle\Phi_2|i_2\rangle$. This also tells us that

$$\mathrm{tr}(\mathbb{E}[\rho_0^{(\mathrm{II})\otimes 2}\hat{Q}\otimes\hat{Q}]\approx 0.$$

We thus see a clear difference in the fluctuations of $\hat{Q}$ for the two protocols.

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
