# Peer review of "Equilibration of quantum cat states"

_SciPost Physics, doi:SciPost Phys. 9, 004 (2020)_

## Round 2 · Referee Report · Anonymous (Referee 1) · 2020-5-16

Report

Jin's work addresses an interesting question, how can one identify cat states after a system has equilibrated? He shows that the time fluctuations in the steady state distinguish cat states from other more traditional states.

The results reported in the paper look technically correct but I think the presentation can be improved. I found some statements that may not be correct. Below I point out some of my concerns and questions in the order they appear in the manuscript. They are not ranked by importance.

1) On page 2, I do not understand this sentence: "We will see that such states present non trivial, possibly non-local fluctuations of the off-diagonal components in the steady-state that are fixed by the initial quantum coherences." The "off-diagonal components" of what?

2) On page 4, what is the "diagonal ensemble"? I also do not understand why the case in which the Hamiltonian is the identity operator ("one energy sector that is the whole Hilbert space") is called the "usual microcanonical ensemble." If the Hamiltonian is the identity operator there is no dynamics so I do not see the point of highlighting that "extreme case".

3) Throughout the manuscript instead of the word "moment" the author incorrectly uses the word "momenta". The first place where I identified this was in the first line on page 5.

4) I suggest the author to number all the equations. Below I have a comment about an equation that is not numbered.

5) I would not call the state for protocol II (unnumbered equation) a "classically mixed state", and I would not call $\hat Q$ an observable. $\hat Q$ is a highly many body operator. Has the author found similar signatures to the ones in the right panels in figure 2 in a local operator?

6) On page 8, what the author describes as the ETH is not what I understand as the ETH. I understand the ETH as a statement about matrix elements of observables that has nothing to do with the initial states. In that sense the discussion in the last paragraph of page 8 does not appear to be an equivalent statement to the ETH but just the ETH applied to the states the author is studying. I should add that $\hat Q$ does not look to me like an operator that would fulfill the ETH.

7) On page 9, the author writes "Typicality states that for all eigenstates of the energy window, few-body operators have thermal distributions in the thermodynamic limit." This is not what I understand as typicality, which I don't think says anything about eigenstates but rather about pure states that are random superpositions of eigenstates in the energy window.

8) Also on page 9 the author writes: "Typically, for finite-size integrable systems for example, one expects the existence of long-lived oscillations that prevents the system from equilibrating [30, 31]." I don't think this is what generally happens in finite-size integrable systems, which I believe equilibrate so long as they are not too small or the initial state is not too special. There is a long literature on this involving the generalized Gibbs ensemble in finite-size integrable systems.

  • validity: good
  • significance: good
  • originality: good
  • clarity: good
  • formatting: good
  • grammar: good

Author:  Tony Jin  on 2020-05-29  [id 842]

(in reply to Report 1 on 2020-05-16)
Category:
answer to question

I thank the referee for very useful comments and criticism. I have not yet made changes appear in the preprint as I am waiting for the editor’s recommendation to do so. Let us address the different points one by one :

1) On page 2, I do not understand this sentence: "We will see that such states present non trivial, possibly non-local fluctuations of the off-diagonal components in the steady-state that are fixed by the initial quantum coherences." The "off-diagonal components" of what?

The sentence may be not precise enough indeed : Here it is implicit that everything refers to the eigenbasis of the Hamiltonian. The off-diagonal components are off-diagonal elements of H. I will add a sentence to clarify this point.

2) On page 4, what is the "diagonal ensemble"? I also do not understand why the case in which the Hamiltonian is the identity operator ("one energy sector that is the whole Hilbert space") is called the "usual microcanonical ensemble." If the Hamiltonian is the identity operator there is no dynamics so I do not see the point of highlighting that "extreme case".

The diagonal ensemble is the ensemble where only the diagonal elements of the density matrices have been kept and the off-diagonal have been set to zero. I will add a sentence to make this definition precise. What I call the usual microcanonical ensemble is the normalized identity matrix written in the Hamiltonian eigenbasis. I highlighted these two cases because it’s an important question in thermalization to know how to go from a system described by the diagonal ensemble to the microcanonical ensemble. What is expected from averaging over time-evolution of ergodic quantum systems is that density matrices are given by the diagonal ensemble. To get that the microcanonical ensemble produces the correct output for the expectation values of observables, one needs additional hypothesis for instance on the distribution of the elements of observables of interest. (See for instance the reference [14] for a discussion of this in the context of random matrix theory and ETH). I discuss how this correspondence can be done in the framework of the paper in the section 4. I will add a sentence to make this motivation clear.

3) Throughout the manuscript instead of the word "moment" the author incorrectly uses the word "momenta". The first place where I identified this was in the first line on page 5.

Thank you for pointing this out. It will be corrected.

4) I suggest the author to number all the equations. Below I have a comment about an equation that is not numbered.

Ok.

5) I would not call the state for protocol II (unnumbered equation) a "classically mixed state", and I would not call Q an observable. Q is a highly many body operator. Has the author found similar signatures to the ones in the right panels in figure 2 in a local operator?

I guess it’s only a semantic problem but here classically mixed state means that protocol II is just a classical statistical superposition of \Psi_1 and \Psi_2 with no quantum coherence. Maybe “mixed state” would be a better terminology. What I call an observable is just a self-adjoint operator in the Hilbert space. Q is indeed a highly non local operator and in general no suited for experimental applications. I will add a note to make this precise. In principle, similar statements concerning the fluctuations hold for local operators as long as they have non zero projection on the off diagonal sector (last line of eq(3)). For instance one could imagine to make an interference experiment between two distant parts of the system. In average the interference should be zero for both protocols but the amplitude of the fluctuations should be greater for protocol I. I will add a few sentences to discuss this local versus non local observables.

6) On page 8, what the author describes as the ETH is not what I understand as the ETH. I understand the ETH as a statement about matrix elements of observables that has nothing to do with the initial states. In that sense the discussion in the last paragraph of page 8 does not appear to be an equivalent statement to the ETH but just the ETH applied to the states the author is studying. I should add that Q does not look to me like an operator that would fulfill the ETH.

I am not sure to see where we disagree, I also state that the ETH is a statement about matrix elements (first paragraph of p 8). I do think though that one has to fix a thin energy shell from which one draws eigenstate upon which expectation values are computed (the energy scale E has to come from somewhere for instance). What I am saying in this part is that we can replace the assumption made in the ETH on the distribution of matrix elements of observables by an assumption on the group upon which the dynamics is invariant. Both assumptions yield the same expectations for observables. I also agree on the fact that Q, by its non-local nature should not fulfill the ETH, hence the need for a more general theory.

7) On page 9, the author writes "Typicality states that for all eigenstates of the energy window, few-body operators have thermal distributions in the thermodynamic limit." This is not what I understand as typicality, which I don't think says anything about eigenstates but rather about pure states that are random superpositions of eigenstates in the energy window.

This is indeed a more accurate statement. It will be corrected.

8) Also on page 9 the author writes: "Typically, for finite-size integrable systems for example, one expects the existence of long-lived oscillations that prevents the system from equilibrating [30, 31]." I don't think this is what generally happens in finite-size integrable systems, which I believe equilibrate so long as they are not too small or the initial state is not too special. There is a long literature on this involving the generalized Gibbs ensemble in finite-size integrable systems.

I think you are correct. I will modify this statement to say that long-lived oscillations can happen even though this may not constitute the most general situation. Also I believe relaxation towards GGE holds for local observables but here I address the equilibrium state of the whole system.

---

## Round 2 · Referee Report · Anonymous (Referee 2) · 2020-6-1

Strengths

1 - Rather fresh and original approach to a very timely problem

2 - The main statements and definitions concisely formulated

3 - Presentation has a very good balance between heuristic presentation and formal derivations and proofs (mainly in appendices)

Weaknesses

1 - Presentation of numerics is a bit "too sketchy" and imprecise

2 - Text needs a bit more of (language) polishing

Report

This is a very nice paper and I strongly recommend its publication. It presents an approach to quantum ergodicity from the viewpoint of fluctuations of physical observables. After a careful thought, the result may not be very surprising, but it is very nicely and clearly formulated. First of all, even the standard definition of quantum ergodicity is given in very general and flexible formulation, allowing for any number of conserved quantities.

The author then generalizes the ergodic-time-aveaging projector to fluctuating observables which is formulated in a tensor product Hilbert space
of n-copies of the original system. Not only the result for the second moment (n=2) is compactly written down, also the general result for any moment order n is derived in appendix. What is remarkable is, that this time-averaging projectors in higher tensor-product spaces are never really ergodic (even for a nicely ergodic system) but they generically keep memory of the initial state. This is demonstrated in a numerical example of XXX chain in a random field, starting from initial cat states.

I particularly find interesting the discussion at the end in relation to integrable systems. Specifically, the presentation as it is formulated refers to systems with finite Hilbert spaces, where one can formulate ergodicity through spectral decomposition. It would be interesting to think of generalizing this to truly extended (infinite) quantum systems, where locality or non-locality of conserved operators would become important.

Requested changes

1 - Discussion of numerical example is a bit imprecise (or sketchy). For example the author refers to "mean eigenvalue spacing" as the `mean ratio of consecutive level spacings'. Please be more precise, as the mean level spacing is an irrelevant quantity.

2 - It is not clear in Figure 2 what is meant by error bars (mentioned in the main text)? I guess there is no "disorder averaging" in numerics, but what is the meaning of the blue-shaded region in the top right panel?

3 - The meaning of operator-absolute value (|Q|) in the last display equation at the end of page 6 is not explained.

4 - In discussion of ETH is section 4 it is not clear why ETH - which is a statement about the distribution of matrix elements of local observables - should depend on the initial state (of course it does through \delta E, in particular when one would want to apply it to cat states where no longer \delta E << E, but I guess this should be explicitly stated).

---

## Round 3 · Referee Report · Anonymous (Referee 1) · 2020-6-21

Report

The author addressed most of my points. The ones not addressed may be considered a matter of interpretation so I recommend publication.

I think by:
"Fully-degenerate spectrum corresponds in general to chaotic or non-integrable systems..."
the author means:
"In general chaotic or non-integrable systems have a fully non-degenerate spectrum..."

I should add that this entire comment seems a bit off-track to me.

---

## Round 3 · Referee Report · Anonymous (Referee 2) · 2020-6-30

Report

The author has addressed appropriately all my comments. I recommend publication of the manuscript in the present form.

---

## Round 3 · Author Response

All comments are made in the list of changes.

---

## Round 3 · List of Changes

I thank both referees again for their very useful comments and criticisms.

Below I make a list of the changes made in the manuscript that address the requests formulated by the referees point by point.

Report 2

1 - Discussion of numerical example is a bit imprecise (or sketchy). For example the author refers to "mean eigenvalue spacing" as the `mean ratio of consecutive level spacings'. Please be more precise, as the mean level spacing is an irrelevant quantity.

Thank you for this precision, this has been corrected.

2 - It is not clear in Figure 2 what is meant by error bars (mentioned in the main text)? I guess there is no "disorder averaging" in numerics, but what is the meaning of the blue-shaded region in the top right panel?

I changed «Details on how these values are obtained and the meaning of the error
bars are given in app.C»
to «Details on how these values and the confidence intervals are obtained are given in app C»
I added a sentence in the caption
«The blue-shaded region in the top-right panel is the consequence of oscillations occurring on a much shorter time scale.»
I do not comment this shorter time scale further because the theory only says something about the amplitude of these fluctuations and not on their dynamical behavior (though it would be something interesting to invest).

3 - The meaning of operator-absolute value (|Q|) in the last display equation at the end of page 6 is not explained.

Q is not the operator, it’s its expectation value (operators are denoted with an hat). |Q| is just the absolute value of Q.

4 - In discussion of ETH is section 4 it is not clear why ETH - which is a statement about the distribution of matrix elements of local observables - should depend on the initial state (of course it does through \delta E, in particular when one would want to apply it to cat states where no longer \delta E << E, but I guess this should be explicitly stated).

Yes, this is exactly the point.

In the discussion, I modified the paragraph

«The important remark is that for the situations we looked at in the
paper, this has no reason to be true anymore, essentially because
we don't expect the off-diagonal correlations to be exponentially
suppressed as there is no notion of ''narrow window'' around a given
energy for the cat states considered.»

into

«In the ETH, the role of the initial state is restricted to fixing the energy scales $E$ and $\Delta E$. The important remark is that ,for the cat states, there is no notion of ''narrow window'' around a given energy anymore, hence we don't expect the ETH to apply. One illustration of that is the fact that off-diagonal correlations are not exponentially suppressed for cat states.»

Report 1

1) On page 2, I do not understand this sentence: "We will see that such states present non trivial, possibly non-local fluctuations of the off-diagonal components in the steady-state that are fixed by the initial quantum coherences." The "off-diagonal components" of what?
The sentence
«we will see that such states present non trivial, possibly non-local fluctuations of the off-diagonal
components in the steady-state that are fixed by the initial quantum coherences.»

has been modified to

«Working in the Hamiltonian basis, we will see that such states present non trivial, possibly non-local fluctuations of the off-diagonal components in the steady-state that are fixed by the initial quantum coherences.»

2) On page 4, what is the "diagonal ensemble"? I also do not understand why the case in which the Hamiltonian is the identity operator ("one energy sector that is the whole Hilbert space") is called the "usual microcanonical ensemble." If the Hamiltonian is the identity operator there is no dynamics so I do not see the point of highlighting that "extreme case".
I modified the whole paragraph to make the motivations more explicit :
«Before going on, let's notice two extreme cases of interest : the first case is when all the energy levels are non-degenerate. Then, $\mathbb{E}[\rho_{0}]$ is just the diagonal ensemble, i.e the density matrix in which one has set all the off-diagonal components to zero. The second case is when there is only one energy sector that is the whole Hilbert space itself. We then have $\mathbb{E}[\rho_{0}]=\frac{1}{d}\mathbb{I}$. In this case, the density matrix tells us that all states with the same energy $E$ have the same probability weight, i.e it is the microcanonical ensemble. Fully-degenerate spectrum corresponds in general to chaotic or non-integrable systems, so one should expect that the diagonal ensemble describes accurately the steady state of such systems \cite{reviewthermalization}. However, in practice, we know that equilibrium states of isolated system are accurately described by the microcanonical ensemble which corresponds to the steady-state of a fully degenerate spectrum. To go from the first ensemble to the second is not a trivial task which requires additional assumptions. We will discuss this point in more details in the section \ref{Discussion}.»

3) Throughout the manuscript instead of the word "moment" the author incorrectly uses the word "momenta". The first place where I identified this was in the first line on page 5.
This has been corrected
4) I suggest the author to number all the equations. Below I have a comment about an equation that is not numbered.
This has been done.
5) I would not call the state for protocol II (unnumbered equation) a "classically mixed state", and I would not call ^Q an observable. ^Q is a highly many body operator. Has the author found similar signatures to the ones in the right panels in figure 2 in a local operator?
I added a few sentences to discuss this :
After introducing \hat{Q}
“Note that $Q$ is non-local, in the sense that it has non zero support on the whole physical space.”
At the end of section 3
“Let us add a remark here. In general because of its high degree of non-locality, it is not expected that $Q$ might be a suitable observable for experimental measurements. But similar qualitative statements about fluctuations should apply for any observables that couple the different energy sectors. For instance, as suggested at the end of \cite{GullansHuse}, one could imagine doing an interference experiment between two parts of the system far part and look at the fluctuations of the pattern.”

6) On page 8, what the author describes as the ETH is not what I understand as the ETH. I understand the ETH as a statement about matrix elements of observables that has nothing to do with the initial states. In that sense the discussion in the last paragraph of page 8 does not appear to be an equivalent statement to the ETH but just the ETH applied to the states the author is studying. I should add that ^Q does not look to me like an operator that would fulfill the ETH.
This meets points 3) of referee 2
In the discussion, I modified the paragraph

«The important remark is that for the situations we looked at in the
paper, this has no reason to be true anymore, essentially because
we don't expect the off-diagonal correlations to be exponentially
suppressed as there is no notion of ''narrow window'' around a given
energy for the cat states considered.»

into

«In the ETH, the role of the initial state is restricted to fixing the energy scales $E$ and $\Delta E$. The important remark is that for the cat states, there is no notion of ''narrow window'' around a given energy anymore, hence we don't expect the ETH to apply. One illustration of that is the fact that off-diagonal correlations are not exponentially suppressed for cat states.»

7) On page 9, the author writes "Typicality states that for all eigenstates of the energy window, few-body operators have thermal distributions in the thermodynamic limit." This is not what I understand as typicality, which I don't think says anything about eigenstates but rather about pure states that are random superpositions of eigenstates in the energy window.
I changed
“Typicality states that for all eigenstates of the energy window, few-body operators have thermal distributions in the thermodynamic limit.”

into

“Typicality states that for all pure states that are random superpositions of eigenstates of the energy window, few-body operators have thermal distributions in the thermodynamic”
limit.

8) Also on page 9 the author writes: "Typically, for finite-size integrable systems for example, one expects the existence of long-lived oscillations that prevents the system from equilibrating [30, 31]." I don't think this is what generally happens in finite-size integrable systems, which I believe equilibrate so long as they are not too small or the initial state is not too special. There is a long literature on this involving the generalized Gibbs ensemble in finite-size integrable systems.

I changed

“Typically, for finite-size integrable systems for example, one expects the existence of long-lived oscillations that prevents the system from equilibrating \cite{XYmodel,BlochBECrevival}.”

into

“For instance, for finite-size integrable systems, there might be long-lived oscillations that prevents the system from equilibrating \cite{XYmodel,BlochBECrevival}.”

---

## Editorial Decision

published